

# Studying the dimensions of m-interactivity and customers' engagement in m-commerce applications

Abdulrahman Abdullah Alghamdi

Department of Computer Science, Shaqra University, Shaqra, Riyadh, Saudi Arabia

## ABSTRACT

These days, technology has a significant role in the success of selling and buying goods through m-commerce applications. Therefore, this study aims to study the impact of dimensions of M-interactivity on customers' engagement in m-commerce and the effect of customers' attention in m-commerce on customer loyalty towards m-commerce applications. The questionnaire was distributed among Saudi citizens. The five hundred questionnaire links were distributed randomly among Saudi citizens. The partial least squares (PLS) were implemented to analyse the data to test the proposed hypotheses and get the rest of the results. The results revealed that the questionnaire used in this study was valid and reliable. Also, it revealed that active control, connectedness, responsiveness, and synchronicity positively impact customers' engagement in m-commerce applications. Furthermore, it showed that ubiquitous connectivity and personalization do not positively impact customers' attention in m-commerce. In addition, the results revealed that customer engagement positively impacts customer loyalty (LO) in m-commerce applications.

Corresponding author
Abdulrahman Abdullah Alghamdi, alghamdia@su.edu.sa

## INTRODUCTION

Technology advances significantly impact societies and businesses (*Alotaibi, 2021*). For example, in a recent report (*Communications, Space & Technology Commission, 2018*), 8 million people use sites and applications to buy goods in Saudi Arabia. Also, the e-commerce market in the kingdom is at least 29.7 billion Saudi riyals with the average online shopping spend by Saudi users being 3.942 riyal (*Communications, Space & Technology Commission, 2018*). In addition, it has been noted that physical stores are being transformed into digital retailers thanks to their interactivity and connection possibilities (*Utami et al., 2022*).

Interactivity refers to a user's capacity to alter the look and feel of a mediated environment in real time. Various feedback techniques can be used online, such as chatbots or moderators, in addition to the ability to modify or edit material and visual display. According to *Roy et al. (2018)*, customer engagement is a subjective assessment of their capacity to participate in the business to meet their purchase objectives. In an online transaction process, interactive elements give consumers greater control over information. Customer relationship management (CRM) and customer engagement are more intimately

related than the purchase process in value co-creation. The ability of mobile e-commerce apps to keep customers engaged *via* interactivity is an important metric to measure the success of such applications. Regarding customer involvement, the importance of interactivity has been overlooked in the past, particularly regarding mobile purchasing tasks (*Alalwan et al., 2020*).

This article suggests the engagement parameters of m-commerce towards customers. The facts and opinions collected from the participants in this survey were studied and a thorough analysis of the data collected by the questionnaire was performed. The data and statistical analysis results provided several facts about customer engagement in m-commerce platforms.

This introduction section has provided a small glimpse of the article's contributions toward customer engagement in m-commerce applications. The 'Literature Review' section provides a detailed description of various factors considered in this study. The section, 'Conceptual Model' represents the conceptual model for this study. It shows the full features and the derivations from this model. The 'Methodology' section depicts the complete methodology applied for deriving results in this study. Finally, the 'Results' section shows tables of results obtained after the statistical analysis. Discussion and conclusion are presented in the last part of this article providing the inferences from the study.

# LITERATURE REVIEW

## Mobile interactivity

Interactivity is the user's ability to modify the mediated environment's form and content in real-time (*Alotaibi & Alshahrani, 2022*). Although previous work has shown a correlation between participation and engagement, more research needs to be conducted. Smartphone device use may be increased through apps' ease of use, interaction, and compatibility (*Kim, Chung & Lee, 2011*). The authors also investigated the association between interactivity and client relationship behavior in live-streaming commerce platforms (*Kim, Chung & Lee, 2011*). They showed that interactivity has a nonlinear effect on consumer participation behavior.

Mobile e-commerce applications provide a different context from previous studies. The authors used a different research setting and found that interaction can increase customer satisfaction and loyalty. Customer engagement represents the dynamic and interactive contact between a consumer and a particular product, positively influencing shopping satisfaction (*Jain, Kaul & Sanyal, 2022*). E-commerce programs may vary from those for other smartphone systems and tools. Indeed, shoppers must purchase and pay for goods and services online rather than *via* an e-commerce program. Furthermore, e-commerce apps must ensure clients feel safe and secure because transactions have inherent risks (*Chan et al., 2022*).

Additionally, interaction is communication. Interactivity and customer engagement were examined in mobile shopping. According to their study, client engagement can be affected by several aspects including active control, customization, universal connectivity, responsiveness, and synchronization. While our current study focuses on the quality

and administration of a mobile e-commerce app, both mentioned authors used different definitions of interactivity. This is crucial as the quality of information and the sense end users have over a mobile e-commerce app significantly influences their willingness to participate in mobile purchasing activities. Based on a literature review, interactivity needs to be unified. Some scholars believe interactivity is a multidimensional construct, while others see interactivity as uni-dimensional. Our study examined six mobile interactivity dimensions: active control, ubiquitous connectivity, connectedness, responsiveness, personalizing, and synchronization. These are described in the following subsections.

### Active control (AC)

Active control (AC) was characterized as a ''user's capacity to actively engage and instrumentally affect an exchange'' (*Liu, 2003*). Many previous studies revealed that AC impacts customers' perceptions and behavior. For example, confidence in electronic commerce would improve if customers completely controlled their online purchasing experiences. Furthermore, consumers' trust and opinions about mobile marketing could be influenced by their ability to actively manage their experience (*Lee, 2005*). Effective planning is vital to consumers' perception of online marketing engagement (*Tan et al., 2018*). In more recent research, customers' willingness to participate in mobile purchasing was positively influenced by active control. Consequently, the following hypothesis is proposed:

H1: Active control (AC) will have a positive impact on customer engagement (CE) in m-commerce.

### Ubiquitous connectivity (UC)

Ubiquitous connectivity (UC) is the user's capacity to access any material, goods, and activities *via* mobile Internet wherever the client wants to. Therefore, customers can shop anywhere as soon as they connect to the Internet thanks to interactive features providing flexibility. In a previous study, customers' impressions and desire to use mobile ticketing technologies were heavily influenced by their familiarity and comfort level with mobile technologies (*Mallat et al., 2008*). Ubiquitous connectivity can have a beneficial effect on customers' willingness to participate in mobile commerce. As a result, the following hypothesis is raised:

H2: Ubiquitous connectivity (UC) will have a positive impact on customer engagement (CE) in m-commerce.

### Connectedness (CN)

Connectedness (CN) refers to the potential of interactive platforms to enable users to be socially engaged and linked with one another. By becoming more social, consumers can discover other people with similar beliefs, experiences, and interests. It has been asserted that customers' satisfaction with social interactions is shaped by their perception and feeling toward people who use the same platforms (*Zhao & Lu, 2012*). Moreover, when customers want to buy items online, they look at other customers' comments about them. This has been considered valuable and reliable for deciding before purchasing. The following hypothesis is thus offered:

H3: Connectedness (CN) will have a positive impact on customer engagement (CE) in m-commerce.

### Responsiveness (RS)

Responsiveness (RS) is linked to customers' perception of the number of responses they received from other customers and marketers to their questions and messages. It has been confirmed that customers need to know about other customers' experiences and that marketers provide appropriate and relevant responses and comparable answers (*Johnson, Bruner II & Kumar, 2006*). It has been noted that customers feeling socially and emotionally related to other customers over an interactive platform yields high responsiveness. The responsiveness positively increases customers' enjoyment and playfulness in m-commerce. Regarding mobile shopping, accessibility has been demonstrated to have a beneficial impact on consumer engagement. As a result, the following theory is proposed:

H4: Responsiveness (RS) will have a positive impact on customer engagement (CE) in m-commerce.

### Personalization (PR)

Making platform features, such as information and design, personalized and compatible with customers' preferences is one of the most important factors in making mobile shopping applications more attractive (*Dholakia, Bagozzi & Pearo, 2004*). It has been asserted that personalization, rather than a mass marketing approach, can enhance users' shopping experience. Furthermore, personalization in social media marketing helps increase customers' intention to purchase (*Li, Kuo & Rusell, 1999*). In recent research, mobile shopping personalization has increased consumer engagement (*Kim & Ko, 2012*). As a result, the following theory is offered:

H5: Personalization (PR) will have a positive impact on customer engagement (CE) in m-commerce.

### Synchronicity (SY)

The concept of synchronicity was described as the amount to which the reaction to a communication event is considered instantaneous or without delay (*Liu & Wei, 2003*). The quality of the communication process shapes customers' satisfaction depending on the time taken to respond to and answer inquiries (*Lee, 2005*). Another study showed that synchrony influences the degree of pleasure people have while utilizing m-commerce (*Yang & Kim, 2012*). Synchronicity has been proven to favor consumer engagement with mobile purchasing. As a result, the following hypothesis is raised:

H6: Synchronicity (SY) will have a positive impact on customer engagement (CE) in m-commerce.

## Engagement behavior

People who interact with goods and services can be called "engaged customers". When it comes to how people engage, several variables are at play. Previous studies have examined how people's attitudes and rewards for becoming involved have changed. Engaging with a business or organization in the digital age means consumers are actively involved

and committed to a brand or organization (*Lin & Wang, 2006*). Emotional, cognitive, and behavioral customer participation were found. Emotional involvement refers to the consumer's happiness and excitement, while behavioral involvement refers to sharing, learning, and promoting the product. Cognitive and behavioral aspects can be separated into four main categories: co-developing, influencing, augmenting, and mobilizing. It is thus possible to enhance and combine the frameworks. Thus, customer interaction may benefit and harm a company's bottom line.

These behaviors constitute consumers' voluntary resource contributions to brands or companies, indicating their active involvement in the digital world, particularly for mobile applications. Customer contributions, such as co-development and behavior enhancement, strive to expand the app's capabilities. Regarding a company's mobile app, such actions can have very different outcomes. Indeed, customers who participate in co-creation can have their ideas approved or rejected by the company. Alternatively, customers participating in improvement may use their methods to increase software without the company's permission.

A lot can be done by convincing and mobilizing. Customer conduct is motivated by a desire to help out fellows and to preserve the mobile shopping applications they use. Changing people's behavior is a customer's desire to spread the word about their apps. Because of their diversity, this wide range of consumer activity necessitates a distinct approach to organizing the predecessors of client engagement for mobile e-commerce applications. Factors such as service fairness, trust, and perceived value for the hotel industry impact all of customer engagement. One of the constructs in a four-dimensional behavior model was a second-order factor (*e.g.*, co-developing, augmenting, influencing, and mobilizing). In this approach, motivational states and probable antecedents are not considered, which can be a significant flaw since customer involvement needs different contributions regarding resources, attention, and buy-in from the leading consumer.

The same goes for organizations that run mobile e-commerce applications, which may want their customers to participate in multiple ways. An organization may prefer co-developing behavior rather than enhancing behavior as it enables them to examine and discuss clients' recommendations for better services on the mobile e-commerce app. For instance, firms might use various strategies to target more specific customer behaviors, such as deconstructing each interaction behavior into its parts.

### Customer engagement (CE)

Using the phrase 'customer engagement', companies, brands, and media can interact with consumers in new and meaningful ways. Loyalty and brand equity may be connected to the value of consumer engagement in finance (sales revenue).

Considering smartphone applications' level of involvement, such as mobile purchases, companies can better attract and keep customers by using these new platforms. Several studies have explored how customers engage with digital marketing campaigns (*Kumar, Purani & Viswanathan, 2018*). The authors seem to have differing views on what 'engagement' means. Customer engagement is how customers interact with companies. The definition of online engagement is one of the most comprehensive: consistently engaging

in brand-related activities on a website or other computer-mediated entity indicates a commitment to the brand's values. Customers and producers can communicate with each other thanks to the interactive nature of mobile buying. Another advantage of mobile shopping is that consumers may contribute to creating value *via* online reviews, ratings, and rankings. However, customers must also maintain a high degree of communication and trade to be actively involved. As a result, clients must be emotionally, behaviorally, cognitively, and socially committed to the interaction process for success. Regarding both aesthetic and practical aspects, as well as monetary ones, several advantages exist. A firm can benefit from increased sales by making these cutting-edge channels available to customers. The current study evaluated customer involvement with m-commerce from three primary perspectives: cognitive, emotional, and behavioral. These aspects of customer connection have often been discussed in the marketing literature. These three characteristics will be considered the second-order determinants of consumer happiness as far as engagement is concerned. Later, we will go further into each of these features.

### Cognitive engagement

The series of persistent and active mental states that consumers feel concerning the principal object of their participation was characterized. Abilities to pay attention and to learn things are among the most critical components of people's overall level of cognitive engagement. Attention is the ability to focus on a specific object of contact. The extent to which a target item (*e.g.*, a product or service) is absorbed in marketing is called absorption; it focuses on one's attention and concern.

### Psychological involvement

One definition of an effective engagement strategy is the summative and persistent degrees of changes a customer feels related to their engagement focus. According to *Dessart, Veloutsou & Morgan-Thomas (2016)*, emotional sub-dimensions of involvement include excitement and pleasure. There is no doubt about it: enthusiasm and joy go hand in hand. Enthusiasm can be defined as a person's inherent desire and eagerness to focus on the subject matter of a relationship. Complementary to pleasure are hedonic consequences, such as joy and amusement, that result from the interaction process with a target item.

### Behavioural engagement

One of the most crucial elements in the engagement process is behavioral engagement. It shows how customers actively participate and connect with brands, organizations, goods, and services. The activation and interaction have been used to define levels of behavioral involvement. A client's involvement with a brand or organization can be determined by the time, energy, and effort they devote to it; it goes further than simply the transaction. There are three main types of social media engagement—sharing, learning, and endorsing—to describe how consumers can share and promote particular brands. This leads to higher customer engagements behavior (*Japutra et al., 2022*)

Increasing client loyalty and satisfaction is not a goal in and of itself; it is technology that companies and brands can use to boost their marketing efficacy. The two main components of customer loyalty are attitudinal and behavioral loyalties, which are

frequently intertwined. Both benefit from three essential parts of consumer engagement: cognitive, emotional, and behavioral. *Harrigan et al. (2017)* presented research in 2017 that effectively verified the predictive impact of social media engagement on customer loyalty. It was shown that consumer engagement was significantly linked to customer loyalty by *Thakur (2016)*. In addition, consumer brand participation is a significant predictor of customer loyalty, as *France, Merrilees & Miller (2016)* demonstrated. Recent research by *Alalwan et al. (2020)* indicated that customer interaction benefits consumer loyalty when it comes to mobile buying. As a result, the following hypothesis is put forward.

## m-commerce loyalty (LO)

E-commerce customer loyalty refers to a customer's positive feelings about a service provider and willingness to promote that service or product to others (*Lee & Wong, 2016*). The desire of a consumer to buy services and goods with the same sentiments throughout time is known as user loyalty, regardless of environmental factors or advertising approaches. Customers' loyalty may be measured in various ways, including the desire to repurchase, customer choices, and positive perceptions of the network operator (*Quach, Thaichon & Jebarajakirthy, 2016*). Mobile loyalty is the desire to return to or to re-use an app for future transactions. Customer loyalty occurs when a customer has a favorable opinion about an e-retailer, resulting in frequent repurchases (*Prashar & Verma, 2020*). Because devoted customers recommend the vendor to their friends and family, the vendor's client base grows, and the company profits and expands. Many firms see customer loyalty as an intangible asset, and, as a result, customer loyalty is a common marketing objective. Customer loyalty can be divided into active and passive (*Massoudi, 2020*). Customer loyalty is a fascinating issue for researchers since it substantially influences a company's competitive advantage. Customer retention is essential in the e-commerce world, and it significantly impacts customers' happiness and revenue (*Brusch, Schwarz & Schmitt, 2019*). Due to the importance of customer loyalty, m-commerce enterprises must establish a favorable relationship with their customers. In other words, the success of m-commerce heavily relies on retailer customer loyalty.

It has been proven in earlier studies (*Yang & Lee, 2017*) that improving customer loyalty and retention is crucial to acquire a competitive advantage. Customer loyalty in m-commerce situations in Taiwan was influenced by perceived value, trust, habit, and customer happiness. In China, consumer loyalty to m-commerce is influenced by perceived security (*Pramesti, 2018*). Regarding Malaysia, efficiency, system availability, fulfillment, privacy, contentment, trust, and commitment influence m-commerce consumer loyalty. The quality of m-commerce service has affected customer loyalty. Customer loyalty to the m-commerce of Zalora in Indonesia is also influenced by trust, satisfaction, and the level of service provided (*Pramesti, 2018*). Some authors have stated that technology alone cannot build customer loyalty in the service industry (*Pereira, de Fátima Salgueiro & Rita, 2016*). They must also consistently give good service to their customers.

H7: Customer engagement (CE) will positively impact customer loyalty towards m-commerce applications.

## CONCEPTUAL MODEL

Our work examined six mobile interaction factors, *i.e.,* active control, ubiquitous connectivity, connectedness, responsiveness, personalization, and synchronization, as shown in Fig. 1. To improve client loyalty, the suggested approach seeks to forecast consumer involvement.

## METHODOLOGY

A questionnaire was used in our research to get responses from people. The target participants were Saudi citizens. The questionnaire had three parts. The first part contained general details about what this study aimed to collect and a process to obtain consent from participants. The study was conducted with the permission of the Research Ethics Committee (REC) at the College of Computing and Information Technology with the reference number: Ethics Appl.220302022. The second part of the questionnaire aimed to gather demographic information. Finally, the third part was used to collect data to analyze the influence of the six mobile interactivity aspects on consumer engagement presented previously. This study used a five-point Likert scale to record answers to questions (5 = Strongly Agree, 4 = Agree, 3 = Neutral, 2 = Disagree, and 1 = Strongly Disagree).

Random sampling was used to generalize the results (*Alotaibi, 2021*). Therefore, we used social networks, WhatsApp groups, and Twitter to disseminate 500 questionnaire links to our intended audience. All 245 completed responses were included.

## RESULTS

### Demographic data

It is clear from Table 1 that most of the respondents (64.5 %) were male. Most participants studied at a level of Bachelor's or Master's (35.1 % and 34.7 %, respectively).

The data used in this study has the following characteristics, as mentioned in Table 2. the data file contains 245 records of information based on various demographic factors. The mean replacement algorithm replaced values with noise or is not at par with the relevance. Smart-PLS (partial least squares) was used to identify and analyze the data. The data metric comprises the mean value of zero and variance of 1. The data was processed over 500 iterations.

### Quality criteria

This section provides the use of instrumental testing steps in this study. First, Smart-PLS was used as the prime methodology tool to evaluate the proposed conceptual model. Second, the indicators, as well as the interactions, were loaded into the model presented. Third, the overall calculation was done with the help of software. Fourth, the hypotheses proposed structural model was loaded into the system. Finally, the inputs from various users collected as a part of the survey were packed into this model for further review.

#### *R square*

R-square is a statistical method that shows the essential proportion of the variance for a variable dependent on a quantity and explained by an independent variable in any

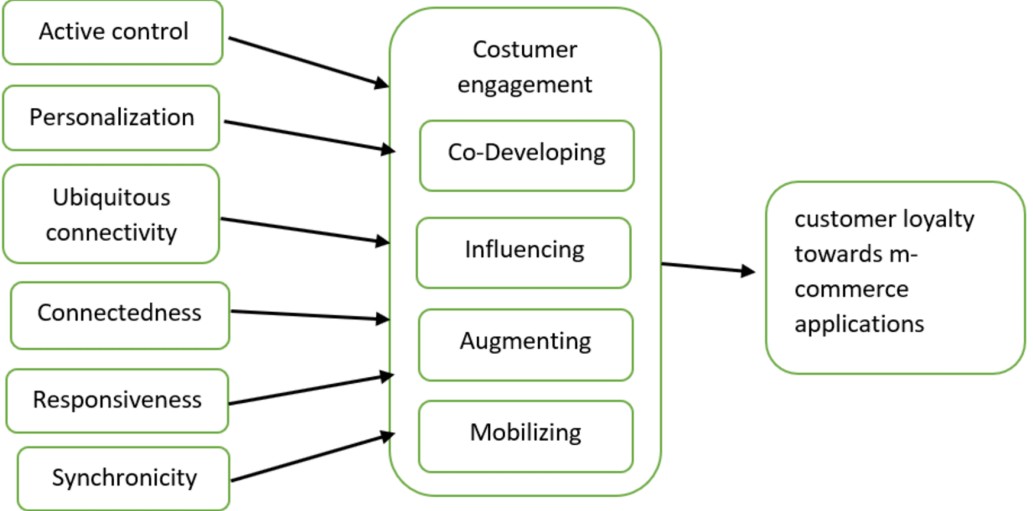

**Figure 1** Illustration of the proposed research model.

**Table 1** Demographic data for participating m-commerce customers.

| Information | Feature | Number of participants | Percentage (%) |
|---|---|---|---|
| Gender | Male | 158 | 64.5 |
| | Female | 87 | 35.5 |
| | Total | 245 | 100 |
| Education level | Diploma | 33 | 13.5 |
| | Bachelor | 86 | 35.1 |
| | Postgraduate | 85 | 34.7 |
| | High school | 41 | 16.7 |
| | Total | 245 | 100 |
| Age | Young | 181 | 73.9 |
| | Old | 64 | 26.1 |
| | Total | 245 | 100 |
| Occupation | Student | 121 | 49.6 |
| | Working in a government organization | 100 | 40.8 |
| | Working in the private sector | 10 | 4.1 |
| | Others | 14 | 5.5 |
| | Total | 245 | 100 |
| Experience | Less than a year | 11 | 4.5 |
| | 1 to less than three years | 114 | 46.5 |
| | Three years or more | 120 | 49 |
| | Total | 245 | 100 |

regression based model. This study comprises various factors represented by variables in the loaded data set. In addition, this data set includes inputs given by multiple recipients who participated in this survey. The R square method was used to identify the proportion

**Table 2  Base data loading for analysis.**

| Data file settings | |
| --- | --- |
| Data file | R1 [245 records] |
| Algorithm to handle missing data | Mean Replacement |
| PLS algorithm settings | |
| Data metric | Mean 0, Var 1 |
| Initial weights | 1 |
| Max. number of iterations | 500 |
| Stop criterion (10-^X): | 7 |
| Use Lohmoeller settings? | No |
| Weighting scheme | Path |
| Construct outer weighting mode settings | |
| AC | Automatic |
| CE | Automatic |
| CN | Automatic |
| LO | Automatic |
| PR | Automatic |
| RS | Automatic |
| SY | Automatic |
| UC | Automatic |

**Table 3  R-square value calculation from the loaded data.**

| Factor | R square | R square adjusted |
| --- | --- | --- |
| CE | 0.627091724 | 0.617690675 |
| LO | 0.474140137 | 0.471976105 |

for these dependent variables under the independent variables calculated for this study, as shown in Table 3 below.

### F-square value

F-square is the change in R-square when an exogenous variable is removed from the model. F-square measures effect size ($\geq 0.02$ is small; $\geq 0.15$ is medium; $\geq 0.35$ is large) (*Cohen, 1988*). Various changes that are reflected in different types of variables under this study were calculated with the help of the f-square value. A couple of factors that were improved by removing certain specific factors are indicated with the help of this value in Table 4.

### Discriminent validity

The discriminant validity for a set of objects is the degree of distinction between the objects for various structures. This validity is one of the most important statistical means for identifying the multiple levels of differentiation between different items and constructs. It is always better to comprise values rather than creating new ideas and associating them with numerous variables. During this analysis, the discriminant measurements were taken

**Table 4   F-square value for the model analysed.**

| Factor | AC | CE | CN | LO | PR | RS | SY | UC |
|--------|----|-----|------|-----|----|----|----|----|
| AC |  | 0.020281077 |  |  |  |  |  |  |
| CE |  |  |  | 0.901647322 |  |  |  |  |
| CN |  | 0.118523461 |  |  |  |  |  |  |
| LO |  |  |  |  |  |  |  |  |
| PR |  | 0.000736959 |  |  |  |  |  |  |
| RS |  | 0.042274161 |  |  |  |  |  |  |
| SY |  | 0.043225334 |  |  |  |  |  |  |
| UC |  | 0.000030452 |  |  |  |  |  |  |

**Table 5   Fornell-Larcker criterion statistics for the study.**

|  | AC | CE | CN | LO | PR | RS | SY | UC |
|----|-------|-------|-------|-------|-------|-------|-------|-------|
| AC | 0.779 |  |  |  |  |  |  |  |
| CE | 0.551 | 0.752 |  |  |  |  |  |  |
| CN | 0.522 | 0.718 | 0.835 |  |  |  |  |  |
| LO | 0.488 | 0.688 | 0.551 | 0.850 |  |  |  |  |
| PR | 0.613 | 0.598 | 0.681 | 0.495 | 0.752 |  |  |  |
| RS | 0.532 | 0.722 | 0.762 | 0.527 | 0.705 | 0.844 |  |  |
| SY | 0.570 | 0.665 | 0.604 | 0.550 | 0.622 | 0.747 | 0.832 |  |
| UC | 0.705 | 0.565 | 0.605 | 0.551 | 0.613 | 0.566 | 0.635 | 0.814 |

**Table 6   Model fit summary for the study.**

| Factor | Saturated model | Estimated model |
|--------|-----------------|-----------------|
| SRMR | 0.068 | 0.071 |
| d_ULS | 6.902 | 7.552 |
| d_G | 3.318 | 3.336 |
| Chi-square | 4,034.781 | 4,051.724 |
| NFI | 0.670 | 0.668 |

care of to identify any overlap in the variance to ensure that a proper correlation existed between each category and the objects that were measured with the help of these categories.

The Fornell and Larcker calculation suggests that the root value of the average variance expected for each variable should be more than the correlation in between these variables. Table 5 suggests the discriminant validity based on these observations.

### *Model fit summary*

Table 6 represents the model fit summary for this study. All the factors under consideration were compared for the saturated model value and the estimated model value.

## Hypothesis testing

To test hypotheses, partial least squares (PLS) were employed to calculate all structural paths between the elements involved. Additionally, the strength of each route was calculated using

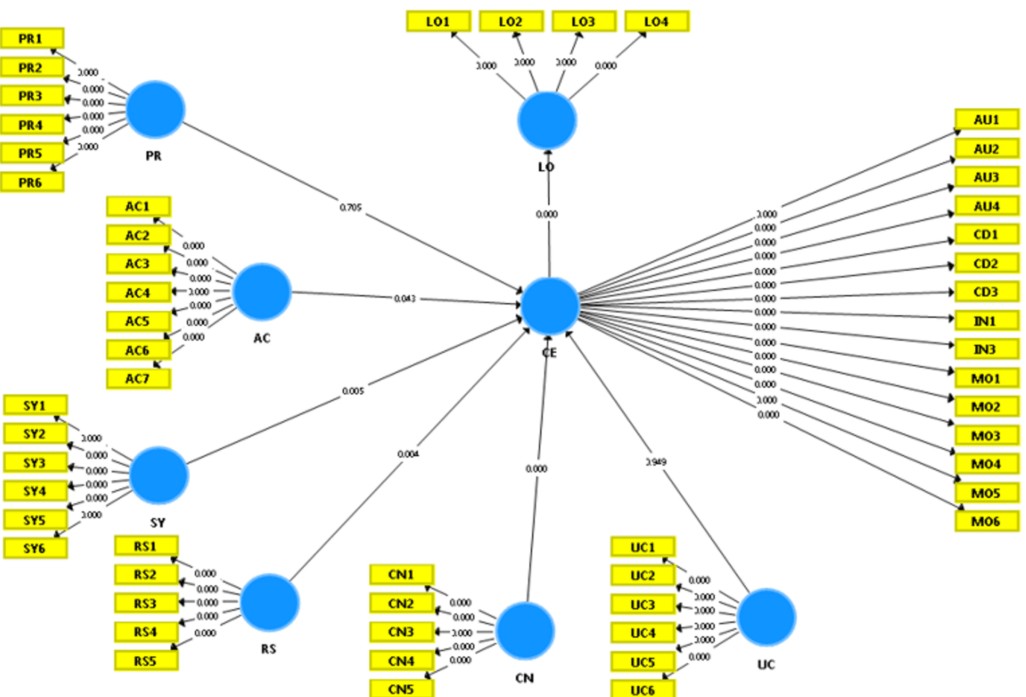

**Figure 2** The bootstrapping results after the data analysis.

a beta value (). Furthermore, a bootstrapping study used T-statistics to evaluate hypotheses (see Fig. 2 and Table 7 below). Active control, responsiveness, and synchrony are shown in Table 4 to enhance consumer engagement positively. According to our findings, customer interaction significantly influences consumer loyalty. On the other hand, personalization and ubiquitous connection seem not to influence consumer engagement substantially.

## DISCUSSION

Based on the results of partial least squares (PLS), active control (AC) has a positive impact on customer engagement (CE) in m-commerce applications. This result is comparable with some previous studies. This result may indicate that customers in m-commerce applications in the Saudi context have complete control over mobile commerce apps. Subsequently, customer engagement with m-commerce applications increased. Therefore, service providers should give customers power when using m-commerce applications.

Contrary to expectations, results showed that ubiquitous connectivity does not positively impact customer engagement in m-commerce applications. This result is incomparable with previous studies. This may indicate that customers need help when they want to connect with mobile commerce apps. This problem may be caused by service providers or the availability of an Internet connection. Service providers should be aware of such issues to increase the use of their m-commerce applications.

Do your facts of the variables calculated with the help of the effect size after the effect size value gives the assessment. In this study, their square value reflects that the conditions

**Table 7  Hypothesis testing result with decision.**

| Factor | Path (hypothesis) | Std. Beta | Std. Error | T-value | *P*-values | Decision |
|--------|-------------------|-----------|------------|---------|-----------|----------|
| H1 | AC - CE | 0.133 | 0.064 | 2.029 | 0.043 | Supported |
| H2 | UC - CE | 0.007 | 0.084 | 0.064 | 0.949 | Not supported |
| H3 | CN - CE | 0.352 | 0.084 | 4.187 | 0 | Supported |
| H4 | RS - CE | 0.244 | 0.084 | 2.9 | 0.004 | Supported |
| H5 | PR - CE | −0.02 | 0.07 | 0.378 | 0.705 | Not supported |
| H6 | SY - CE | 0.205 | 0.074 | 2.814 | 0.005 | Supported |
| H7 | CE - LO | 0.692 | 0.049 | 14.144 | 0 | Supported |

**Table 8  Effect size calculation for the factors.**

| Variables | AC | CE | CN | LO | PR | RS | SY | UC |
|-----------|----|----|----|----|----|----|----|----|
| AC | | 0.130517802 | | 0.089871732 | | | | |
| CE | | | | 0.688578345 | | | | |
| CN | | 0.353447573 | | 0.243376344 | | | | |
| LO | | | | | | | | |
| PR | | −0.0266049 | | −0.018319558 | | | | |
| RS | | 0.242239951 | | 0.166801185 | | | | |
| SY | | 0.20931817 | | 0.144131959 | | | | |
| UC | | 0.005423336 | | 0.003734391 | | | | |

for all the exogenous factors are acceptable within limits as per the observations in Table 8. Multiple regression is possible for identifying the components of different variables and their impacts on each other. All the values below the range of 0.02 and 0.15 have a small impact. However, the values above 0.15 and 0.36 have a medium effect. Finally, the values that are above 0.36 have a higher impact value.

This study's reliability was checked using Cronbach's alpha and composite reliability. It has been suggested that the latter should be above 0.7 (*O'Hair, Friedrich & Dixon, 1998*). However, as can be seen in Table 9, Cronbach's alpha was more than 0.70, and the validity scales were greater than 0.7 for all factors. In addition, in this study, convergent validity was checked using the average variance extracted (AVE) value which must be more than 0.5 to get an acceptable level of convergent validity. All factors' AVE values were higher than 0.5, which indicates that the factors' convergent validity is confirmed.

As expected, results showed that connectedness positively impacts customer engagement in m-commerce applications. This result is comparable with previous studies. Mobile commerce apps may be more successful if users share their experiences with others who use them, which may increase app usage. Additionally, they will be more inclined to stay loyal to the brand they already like. Therefore, service providers should be aware of this factor to increase the use of their m-commerce applications.

Furthermore, results showed that responsiveness positively impacts customer engagement, as shown before. This finding may suggest that consumers' interest in mobile commerce apps rises when they get many replies to their queries and comments.

**Table 9 Analysis of results from the conceptual model.**

| Factor | Cronbach's alpha | Composite reliability | The average variance extracted (AVE) |
|---|---|---|---|
| AC | 0.891 | 0.915 | 0.607 |
| CE | 0.945 | 0.951 | 0.566 |
| CN | 0.891 | 0.92 | 0.697 |
| LO | 0.872 | 0.913 | 0.723 |
| PR | 0.846 | 0.887 | 0.566 |
| RS | 0.899 | 0.926 | 0.713 |
| SY | 0.912 | 0.931 | 0.694 |
| UC | 0.897 | 0.922 | 0.664 |

This result also shows that m-commerce apps can answer unique inquiries and engage highly with clients.

Yet results showed that personalization does not positively impact customer engagement. This result is incompatible with those already obtained. This may indicate that m-commerce apps do not offer customized information searches to customers or that customers need to bout this factor when using m-commerce apps. Thus, service providers should be aware of this factor.

Finally, results showed that synchronicity positively impacts customer engagement. This may indicate that customers are happy with services that quickly respond to requests, process their input, and provide information. As expected, customer engagement positively impacts customer loyalty in m-commerce applications, *i.e.,* when customers' attention to m-commerce apps increases, this leads to commitment towards m-commerce applications.

## CONCLUSIONS

This study aimed to study the impact of m-interactivity dimensions on customers' engagement in m-commerce applications and the effect of customers' attention in m-commerce on customer loyalty towards m-commerce applications. First, it was shown that the instrument used in this study was reliable and valid. Also, the results revealed that active control, connectedness, responsiveness, and synchronicity positively impact customers' engagement in m-commerce applications. Furthermore, it was shown that ubiquitous connectivity and personalization do not positively impact customers' engagement in m-commerce applications. In addition, the results revealed that customer engagement positively impacts customer loyalty in m-commerce applications.

This study may contribute to the theory by filling a gap in the literature in an m-commerce context through empirical results on the impact of m-interactivity dimensions on customer engagement in m-commerce and the effect of customers' attention in m-commerce on customer loyalty towards m-commerce applications. Also, this study may contribute on the practical side by providing a clear picture of the impact of m-interactivity dimensions and customers' engagement in m-commerce applications to assist retailers and service providers in implementing m-commerce applications effectively in the Saudi context.

### Funding

This work was supported by the Deanship of Scientific Research at Shaqra University. The funders had no role in study design, data collection and analysis, decision to publish, or preparation of the manuscript.

### Grant Disclosures

The following grant information was disclosed by the author:
The Deanship of Scientific Research at Shaqra University.

### Competing Interests

The author declares that there are no competing interests.

### Author Contributions

- Abdulrahman Abdullah Alghamdi conceived and designed the experiments, performed the experiments, analyzed the data, performed the computation work, prepared figures and/or tables, authored or reviewed drafts of the article, and approved the final draft.

### Ethics

The following information was supplied relating to ethical approvals (i.e., approving body and any reference numbers):

The Research Ethics Committee (REC) at the College of Computing and Information Technology, Shaqra University granted the approval to conduct this study (Ref. No: Ethics Appl.220302022).

### Data Availability

The raw data is available in the Supplemental File.

### Supplemental Information

Supplemental information for this article can be found online at http://dx.doi.org/10.7717/peerj-cs.1392#supplemental-information.

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
