# Peer review of "Studying the dimensions of m-interactivity and customers’ engagement in m-commerce applications"

_PeerJ Computer Science, doi:10.7717/peerj-cs.1392_

## Round 0.1 · original submission · Minor Revisions

You are advised to prepare the revised version of your paper, keeping in view the suggestions of the experts and resubmit according to their instructions.

Reviewer 1 ·

Basic reporting

- In this paper, the authors proposed a study to test the impact of dimensions of M-Interactivity on customers’ engagement in m-commerce and the effect of customers’ attention in m-commerce on customer loyalty towards m-commerce applications. The idea is good and could be beneficial for the commerce community in general. There are some improvement suggestions as follows

- The use of ‘M’ and ‘m’ for multidimensionality may be consistent throughout the manuscript.
- Figure 2 is not cited in the manuscript.
- Tables 3, 6, 7, and 8 are not cited.
- The author must describe the contribution in the introduction section.
- The author must include a paragraph at the end of the introduction section describing the paper's layout.
- Before the citation check, the author used double inverted commas ‘ ” ’ at many places.
- Use the inverted commas correctly, such as on page 3, line 119,”engaged customers,” should be “engaged customers”.
- On page 3, line 146, clients’ recommendations should be client’s recommendations.
- On page 8, line no. 293, the authors write, “see the figure and table below” the authors must cite the figure and table.
- There are many grammatical mistakes present in the paper. The paper must be proofread very carefully.

Experimental design

- The authors assigned different values to experiment parameters but did not explain why they used these values.

Validity of the findings

- The author used different statistical methods to validate the results but did not explain the results given in Tables 3 to 6. Presenting the results (What mean of these values) is recommended.

Additional comments

- Included in Basic reporting.

Reviewer 2 ·

Basic reporting

This paper presents a study related to customer engagement in m-commerce applications. A questionnaire sample of 500 users was distributed. The acquired results were analyzed using the Partial least squares (PLS) technique. For the improvement of the work following are my suggestions:
• Introduction Section is too brief. The authors should extend the introduction section and add motivation-related discussion.
• Related work section needs an extension. Add more papers relevant to the topic. The literature should be recent from reputed Journal/conference venues.
o A table of comparison of these related approaches should be highly beneficial.
• Section 2.1.5 Personalization (PR) and 2.1.6 Synchronicity (SY) do not refer to any existing related study. Add appropriate references.
• Authors mention that they examine 6 mobile interaction factors. The rationale of using these 6 interaction factors is not clear. Add a related discussion on why only these 6 interactions were considered for the proposed model.
• Figures and Table's names should be in sentence case. For example, the caption of Table 1 needs correction.
• Interpretation of results depicted in Table 3. R-Square Values were not discussed. Add relevant discussion.
• Table 4 has many fields empty, these must be filled. For those fields where a valid value does not exist the terms N/A or “-“ symbol could be inserted.
• Table 6 is not referred to in the paper text. Refer to it and discuss it accordingly.
• There paper needs a full revision of the text to improve the language.

Experimental design

Comments have been mentioned in the basic reporting form.

Validity of the findings

Comments have been mentioned in the basic reporting form.

Additional comments

Comments have been mentioned in the basic reporting form.

---

## Round 0.2 · accepted · Accept

Congratulations on the acceptance of your paper.

Reviewer 1 ·

Basic reporting

The authors incorporated all the given suggestions.

Experimental design

The authors incorporated all the given suggestions.

Validity of the findings

The authors incorporated all the given suggestions.

Reviewer 2 ·

Basic reporting

The revisions are satisfactory and the manuscript can be accepted now.

Experimental design

The revisions are satisfactory and the manuscript can be accepted now.

Validity of the findings

The revisions are satisfactory and the manuscript can be accepted now.

Additional comments

No further comments.